# Subsequent primary cancer incidence among cancer survivors in the United States, 1975–2019: An age–period–cohort analysis

Hui G. Cheng[1]*, Livingstone Aduse-Poku[1], Chelsey McGill[1,2], Oxana Palesh[1,2‡], Susan Hong[1,3‡]

1 Cancer Prevention and Survivorship Outcomes Research Lab, Massey Comprehensive Cancer Center, Virginia Commonwealth University, Richmond, Virginia, United States of America, 2 Department of Psychiatry, Virginia Commonwealth University School of Medicine, Richmond, Virginia, United States of America, 3 Division of Hematology, Oncology and Palliative Care, Department of Internal Medicine, Virginia Commonwealth University School of Medicine, Richmond, Virginia, United States of America

‡ These authors are co-senior authors on this work.
* hcheng3@vcu.edu

## Abstract

### Background

The growing population of cancer survivors faces elevated risks of subsequent primary cancers (SPCs), yet temporal patterns in SPC incidence remain poorly understood. This study aims to characterize age-, period-, and cohort-specific patterns in SPC incidence among US cancer survivors using population-based data.

### Methods and findings

We conducted a retrospective cohort study using Surveillance, Epidemiology, and End Results (SEER) 8 registries, identifying 3.36 million individuals diagnosed with a first primary cancer between 1975 and 2019. Survivors were followed through 2022 to estimate the incidence of SPCs. We used age–period–cohort analysis to estimate longitudinal age curves, cohort and period rate ratios, and annual percent changes in SPC incidence. Analyses were stratified by sex and the five most common index cancer sites. During 29.5 million person-years of follow-up, 510,340 SPCs were observed. SPC incidence increased with age at index cancer diagnosis, rising among females from 915 per 100,000 person-years at ages 35–39 years to 1,980 per 100,000 at ages 75–79 years, and among males from 1,228 per 100,000 to 2,945 per 100,000 across the same age groups, demonstrating steeper rises in men. Cohort-specific SPC risk peaked in the 1935–1945 birth cohorts and declined in later cohorts, except among female survivors of lung cancer and male survivors of bladder cancer, where risks continued to rise. Period patterns showed overall declines in SPC incidence, particularly among survivors diagnosed at a younger age, but increasing risks among survivors diagnosed at an older age and survivors of specific index cancer sites. Notably,

**Data availability statement:** The data underlying this article (SEER 8 registries, 2024 submission) are available through National Cancer Institute Surveillance Epidemiology, End Results Program via SEER*Stat software https://seer.cancer.gov/help/seerstat. Incidence data (S1 Table) and age–period–cohort model estimates (S2 Table) are provided in Supporting information.

**Funding:** This work was supported by the US National Cancer Institute (R01CA239714, R01CA172145, and R01CA226080 to OP). Additional support was provided by the National Cancer Institute (P30CA016059 to the Massey Comprehensive Cancer Center), which supplied infrastructure and shared resources used by the authors in conducting this study. The funding agencies and sponsors had no role in the design or conduct of the study; in the collection, management, analysis, or interpretation of the data; in the preparation, review, or approval of the manuscript; or in the decision to submit the manuscript for publication.

**Competing interests:** I have read the journal's policy and the authors of this manuscript have the following competing interests: O.P. serves as a consultant for Brigham Young University; University of Rochester; Shook, Hardy and Bacon; Elsevier; Merck; NIH; Sage Publisher; and Monash University. Other authors declare no conflict of interest relevant to this manuscript.

**Abbreviations:** IRR, incidence rate ratio; SEER, Surveillance, Epidemiology, and End Results; SPCs, subsequent primary cancers; STROBE, Strengthening the Reporting of Observational Studies in Epidemiology.

SPC incidence rose by 60% among female lung cancer survivors between 1975–1979 and 2015–2019 (incidence rate ratio = 1.60, 95% CI [1.22, 2.09]; $p < 0.001$). Main limitations include the descriptive nature of age–period–cohort analyses and the absence of treatment, genetic, and lifestyle data in SEER.

## Conclusions

SPC risk is shaped by complex, site- and sex-specific temporal patterns. These findings underscore the need for tailored survivorship care strategies that incorporate age, cohort, and index cancer site to mitigate future SPC burden.

### Author summary

#### Why was this study done?

- The cancer survivor population is growing, and they remain at risk of developing new cancers.
- Understanding how these risks change over time and across generations can help guide prevention and monitoring.
- We used data from over 3 million cancer survivors in the US to study patterns of subsequent primary cancers (SPCs).

#### What did the researchers do and find?

- We analyzed cancer registry data in the United States from 1975 to 2021 to examine age, calendar period, and birth cohort effects on SPC risk.
- We found that SPC risks varied by cancer type, sex, and generation, with complex patterns across time.
- SPC risks generally increased with older age at first cancer diagnosis, but breast cancer survivors showed stable risks across ages.
- With a few exceptions (such as female lung cancer and male bladder cancer survivors), SPC risks declined in more recent decades; however, survivors diagnosed at older ages continued to face persistent or rising risks.

#### What do these findings mean?

- These results highlight the importance of long-term monitoring of cancer survivors, as risks of SPCs remain substantial.
- The findings can inform prevention strategies, survivorship care, and public health planning.
- A main limitation is that we do not know the specific causes of the SPC observed. Future studies are needed to understand why these patterns occur.

## Introduction

The global population of cancer survivors is rapidly expanding, driven by aging demographics and advances in early detection and treatment [1,2]. Worldwide, cancer cases are projected to rise 77% between 2022 and 2050 [1]. In the United States (US), the survivor population is expected to grow by 22% over the next decade—from 18 million in 2025 to more than 22 million by 2035 [2]. This growing cancer survivor population faces unique long-term health challenges that extend beyond acute treatment outcomes.

One critical concern is the elevated risk of developing subsequent primary cancers (SPCs), distinct malignancies arising in different anatomical sites or with different histological characteristics in the same individual [3]. Cancer survivors are at increased risk for SPCs due to a combination of aging, treatment-related exposures (e.g., radiation, chemotherapy), genetic predisposition, and persistent lifestyle factors such as tobacco smoking, alcohol drinking, poor diet, obesity, and physical inactivity [4–13].

Recent studies have documented rising trends in SPC incidence over the past four decades [13,14]. For example, an analysis of Surveillance, Epidemiology, and End Results (SEER) 1973–2000 data found increases in the standardized incidence ratio of SPC during the last five years of the study period, following two decades of stable trends. These patterns may reflect shifts in population aging, medical practices, and generational exposures.

Age–period–cohort modeling offers a robust framework to decompose these influences by separating the effects of biological aging (age), temporal changes affecting all individuals (e.g., improvements in screening or treatment; period), and generational differences in exposures and behaviors (such as smoking; cohort). This approach has been widely applied to cancer incidence and mortality and provided valuable insights [15–17], but to our knowledge, has not been used to examine SPC incidence in the US using population-based data.

In this study, we apply age–period–cohort modeling to evaluate temporal patterns in SPC incidence among US men and women diagnosed with an index cancer between 1975 and 2019, using data from the SEER program. We stratify analyses by the index cancer site to assess site-specific patterns that may inform personalized surveillance and survivorship care. Our goal is to advance understanding of SPC etiology and contribute to efforts aimed at improving quality of life and reducing cancer burden among survivors.

## Methods

### Study design and data source

In this retrospective cohort study, we evaluated the incidence of SPCs among individuals diagnosed with a first primary cancer between 1975 and 2019, followed through December 2022, using data from the SEER (Research Resource Identifier [RRID]: SCR_006902) 8 registries (2024 submission), which cover approximately 10% of the US population [18].

SPCs were defined according to SEER's multiple primary rules [19]. Following SEER's recommendations, we excluded cases diagnosed at autopsy or via death certificate only, and restricted follow-up to survivors with at least two months post-index cancer diagnosis to reduce misclassification of synchronous primaries [20].

### Statistical analysis

We first obtained the number of SPC events and person-years at risk (from index cancer diagnosis to death or end of follow-up) for each age and year stratum by conducting a Multiple Primary-Standardized Incidence Ratio (MR-SIR) session in SEER*Stat software [21]. SPC incidence was calculated as the number of individuals diagnosed with an SPC divided by total person-time at risk. For males and females diagnosed with an index cancer, we estimated SPC incidence across 5-year calendar periods of index cancer diagnosis (e.g., 1975–1979, 1980–1984) and 5-year age groups (e.g., 25–29 years, 30–34 years). Nominal birth cohorts (e.g., 1,930 cohort) were derived by subtracting mid-year age (e.g., 57 for 55–59 years) from mid-year period (e.g., 1987 for 1985–1989), resulting in partially overlapping cohorts referenced by mid-year of birth. The earliest calendar year served as the reference period.

We then used weighted least squares age–period–cohort models, developed by Rosenberg and colleagues [22], to decompose observed incidence into:

a. **Age effects:** risks associated with biological aging

b. **Period effects:** calendar-year patterns

c. **Cohort effects:** risk differences across birth cohorts such as generational exposures leading to varying risks of developing SPCs for individuals of the same generation. Cohort effects were modeled as interactions between age and period. In this study, age–period–cohort analyses were conducted using the Rosenberg method [23]. The Rosenberg approach resolves the identifiability problem (i.e., age, period, and cohort effects are linearly dependent) by estimating uniquely defined functions of the parameters. These include net drift (overall annual percentage change), longitudinal age patterns, and period and cohort deviations. Linear trends are absorbed into the net drift, while deviations are constrained to sum to zero and decomposed into orthogonal components, ensuring interpretability. Cohort deviations are weighted to account for the variable number of periods each cohort is observed. Parameters were estimated using weighted least squares under the assumption that SPC counts follow a Poisson distribution. Person-years at risk were included via a log offset to account for varying follow-up times, and extra-Poisson variation was incorporated to account for overdispersion in the data [22,23].

We first generated **longitudinal age curves**, which trace cohort-specific SPC incidence across age groups. These curves differ from cross-sectional age curves, which reflect SPC incidence among older cohorts at older ages and younger cohorts at younger ages. When cohort effects are present, cross-sectional curves may artifactually suggest descending or ascending SPC incidence with age. In contrast, longitudinal curves provide a more accurate depiction of aging-related processes by accounting for cohort and period shifts [24]. We then generated **period rate ratio curves** and **cohort rate ratio curves**, which are composite curves that represent the incidence rate ratio (IRR) of each period or cohort relative to the reference period or cohort. Last, we estimated age-specific **annual percent change** (local drift) as well as the overall average **annual percent change** (i.e., net drift) to quantify temporal shifts of SPC incidence.

We conducted age–period–cohort analyses for the five most common index cancer sites among females (i.e., breast, lung and bronchus, colon and rectum, uterine corpus, and melanoma of the skin) and males (i.e., prostate, lung and bronchus, colon and rectum, urinary bladder, and melanoma of the skin) using stratified analysis, with each site analyzed independently [25].

Analyses were conducted using SEER*Stat (version 9.0.33.0) [21] and the National Cancer Institute Age-Period-Cohort Web Tool [22] (https://analysistools.cancer.gov/apc/; last accessed 12/05/2025), both are publicly available. Statistical uncertainty was quantified by 95% confidence intervals. Results were visualized using Stata 18.0. The study used publicly available, de-identified SEER data and does not require institutional review board oversight. Analyses were planned from February to August 2025. This study is reported as per the Strengthening the Reporting of Observational Studies in Epidemiology (STROBE) guideline (S1 STROBE Checklist). Incidence data and age–period–cohort model estimates are provided in Supporting information (S1 Table).

## Results

The sample included 3,363,683 cancer survivors diagnosed with an index primary cancer between 1975 and 2019, comprising 1,646,256 women (49%) and 1,717,427 men (51%). During 29,539,667 person-years of follow-up, a total of 510,340 SPCs were observed—235,099 among women and 275,241 among men. Among women, there were 539,378 survivors of breast cancer, 174,523 survivors of colon and rectum cancers, 152,731 survivors of lung and bronchus cancers, 117,545 survivors of uterine corpus cancer, and 68,199 survivors of skin melanoma. Among men, there were 520,948 survivors of prostate cancer, 205,609 survivors of lung and bronchus cancers, 182,088 survivors of colon and

rectum cancers, 108,586 survivors of urinary bladder cancer, and 60,427 survivors of skin melanoma. Collectively, the five most common index cancer sites accounted for 64% of female survivors and 63% of male survivors in the sample.

## Longitudinal age-specific SPC incidence

When all sites were considered, SPC incidence increased with age at diagnosis of the index cancer, independent of period and cohort effects among both men and women (Fig 1A). Incidence rates were nearly identical between sexes when the index cancer was diagnosed before age 20 years. Beyond age 20 years, SPC incidence rose in a linear fashion (e.g., SPC incidence rose from 915 per 100,000 person-years at ages 35–39 years to 1,980 per 100,000 at ages 75–79 years among females, and from 1,228 per 100,000 to 2,945 per 100,000 across the same age groups among males), with a steeper increase among men (compared to women). After age 80 years, incidence began to decline in women, whereas it plateaued in men.

When stratified by the site of the index cancer, the general increasing incidence with age was observed among survivors of four of the five most common index cancer sites among women (lung/bronchus, colon/rectum, uterine corpus, melanoma) and all five most common index cancer sites among men (prostate, lung/bronchus, colon/rectum, bladder,

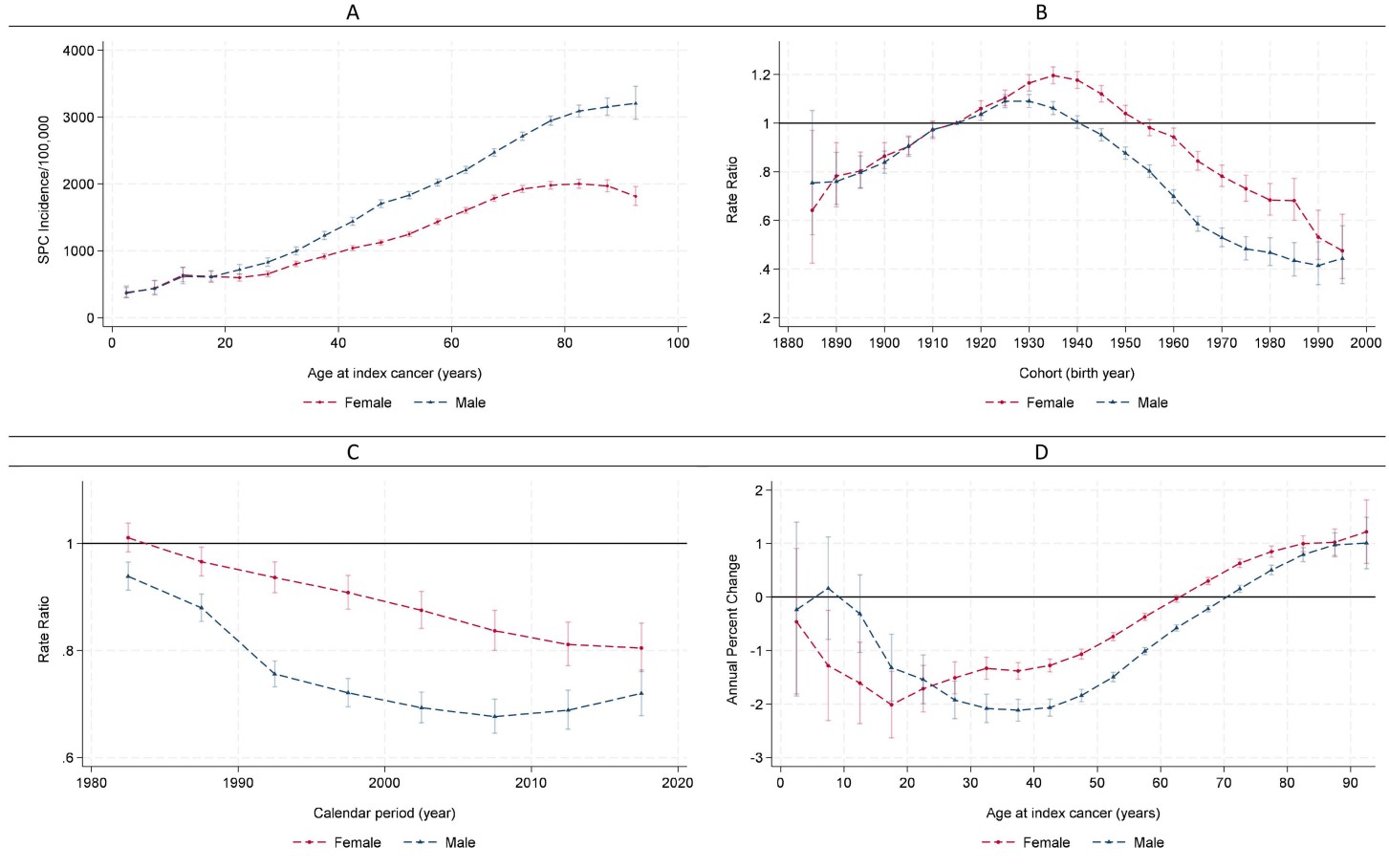

**Fig 1. Estimated age, period, cohort effects of subsequent primary cancer (SPC) by sex.** Data from SEER 1975-2019. **(A)** Longitudinal age-specific incidence of SPCs. **(B)** Cohort incidence ratio ratios (reference cohort = 1,915). **(C)** Period incidence rate ratios (reference period = 1975-1979). **(D)** Age-specific annual percent change. Verticle bars represent 95% confidence intervals.

melanoma) (Fig 2). An exception was among female breast cancer survivors, where SPC incidence showed much slighter variations with a nadir of 1,328 per 100,000 person-years when diagnosed at ages 40–45 years. Across sites of the index cancer, uterine corpus cancer survivors had the lowest SPC incidence among women diagnosed with the index cancer between the age of 40 and 80 years, and prostate cancer survivors had the lowest incidence among men across all ages. Among those whose index cancer was diagnosed before age 40 years, breast cancer survivors had the highest SPC incidence among women, whereas colorectal cancer survivors had the highest incidence among men. Among those whose index cancer was diagnosed after age 70 years, survivors of skin melanoma had the highest risk of SPC in both men and

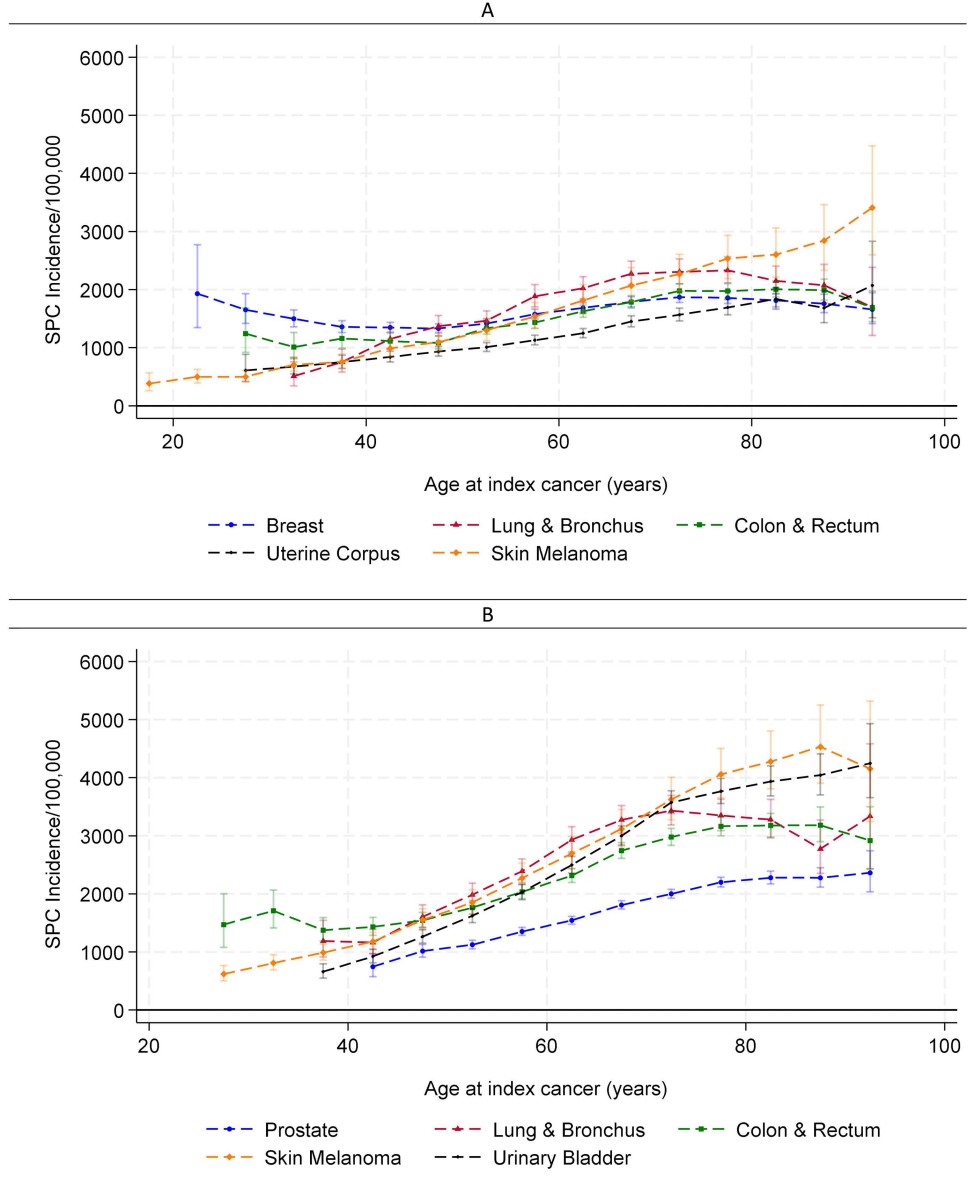

**Fig 2. Longitudinal age-specific incidence of subsequent primary cancer (SPC) by site of index primary cancer.** Data from SEER 1975–2019. **(A)** Among females. **(B)** Among males. Verticle bars represent 95% confidence intervals.

women. Among survivors of lung cancer, colorectal cancer, and skin melanoma, men tended to have higher SPC risks than women.

Observed age-specific incidence and fitted longitudinal curves were shown in Fig A (males) and Fig B (females) in S1 Supplemental Figures. The parallel shapes suggest that the age–period–cohort model adequately captured the main features of the data.

### Cohort-specific patterns

When all sites are considered, the risks of SPC for cohorts born between 1885 and 1995 peaked at the 1925 and 1930 cohorts among men and the 1,935 cohort among women and declined in later cohorts. Among men, the decline after the peak slowed down in the most recent cohorts starting in the 1,970 cohort; among women, the decline continued with a plateau in the 1,985 cohort (Fig 1B).

Cohort-specific risks of SPC varied by the index cancer site (Fig 3). Among women survivors of lung and bronchus cancer, SPC risk increased across successive cohorts, peaked in the 1,940 cohort, and stabilized in cohorts born thereafter (Fig 3A). Among breast cancer survivors, SPC risk remained stable through the 1,940 cohort, followed by a marked decline in later cohorts. Risks of SPC among survivors of colorectal and uterine corpus cancer followed trajectories that rose in earlier cohorts, peaked among cohorts born between 1935 and 1945, and then declined in more recent cohorts. Among men, SPC incidence increased across successive cohorts of urinary bladder cancer survivors (Fig 3B). Prostate and colorectal cancer survivors demonstrated trajectories with peaks in the 1925–1930 cohorts. SPC risks among lung and melanoma survivors peaked later, around 1940–1950. Of note, modest upticks in SPC risk were observed for survivors of skin melanoma (male and female), uterine corpus cancer (female), and prostate cancer (male) in the most recent cohorts, although estimates were less precise.

### Period patterns

When all sites are considered, the risk of SPC generally declined over calendar periods with larger declines among male cancer survivors (annual percent change = −0.63%, 95% CI [−0.76%, −0.49%]; $p < 0.01$ among women and annual percent change = −0.95%, 95% CI [−1.09%, −0.81%]; $p < 0.01$ among men; Table 1). Among women, the decline was fairly linear (Fig 1C). Among men, the decline plateaued during 2005–2009 and then showed a slight uptick in the two more recent periods. The declines were mostly driven by decreasing SPC risks (indicated by negative annual percent changes; Fig 1D) among female cancer survivors diagnosed with the index cancer between the age of 5 and 60 years and male cancer survivors diagnosed between the age of 15 and 70 years. Among survivors diagnosed at an older age, the risk of SPC increased.

Nonetheless, patterns varied by the site of the index cancer. Among female cancer survivors, SPC incidence increased over time following an initial diagnosis of lung and bronchus cancer (annual percent change = 1.12%, 95% CI [0.53%, 1.85%]; $p < 0.001$), leading to a 60% higher incidence observed in the 2015–2019 calendar period compared to 1975–1979 (IRR = 1.60, 95% CI [1.22, 2.09]; $p < 0.001$; Fig 4A). Age-specific annual percent change revealed steeper increases among survivors diagnosed with the index cancer after age 50 years (Fig 4C).

In contrast, SPC incidence among breast cancer survivors remained stable from 1980 to 2000, followed by a progressive decline, culminating in a 29% reduction in 2015–2019 relative to 1975–1979 (IRR = 0.71, 95% CI [0.65, 0.78]; $p < 0.001$). This decline was primarily driven by decreases among survivors diagnosed before age 70 years (Fig 4A and 4C).

Among female survivors of colorectal cancer, SPC incidence declined between 1980 and 2000 and subsequently plateaued. For survivors of skin melanoma, a modest increase in SPC incidence was observed after 2005, with a 15%−20% rise across the three most recent calendar periods (Fig 4A).

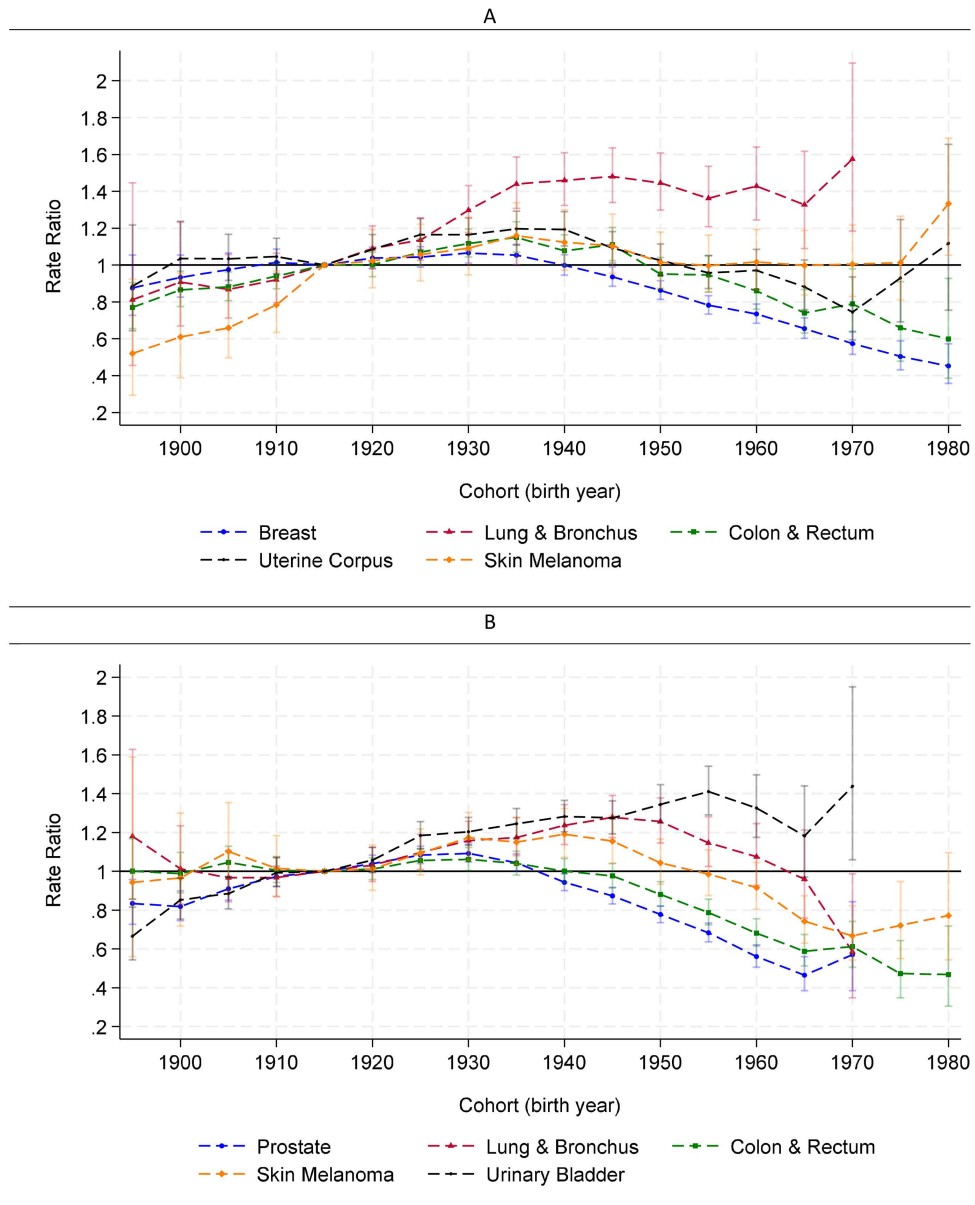

**Fig 3. Estimated cohort effect (rate ratio) of subsequent primary cancer (SPC) by site of index primary cancer.** Data from SEER 1975–2019. **(A)** Among females. **(B)** Among males. Verticle bars represent 95% confidence intervals. Reference cohort = 1,915. A rate ratio >1 indicates higher incidence of SPC among cancer survivors of a given birth cohort compared to the reference cohort and vice versa. Estimates for cohort later than 1975 were not presented for survivors of lung cancer, bladder cancer, and prostate cancer due to imprecision.

Among male cancer survivors, SPC incidence declined over time following diagnoses of colorectal cancer, prostate cancer, and skin melanoma. In contrast, a slight increase was observed among survivors of bladder cancer (Table 1 and Fig 4B).

Age-specific annual percent change revealed a consistent pattern across cancer types: lower annual percent changes (often negative indicating a declining trend in SPC incidence) among survivors diagnosed with the index cancer at younger ages, and higher annual percent changes (often positive indicating increasing SPC incidence) among those

**Table 1. Estimated annual percent change of subsequent primary cancer incidence among individuals diagnosed with an index cancer. Data from surveillance, epidemiology, and end results 1975–2019.**

| Site of index cancer | Female | | | Male | | |
|---|---|---|---|---|---|---|
| | **Annual percent change** | **95% CI** | ***p*-value** | **Annual percent change** | **95% CI** | ***p*-value** |
| All sites | **−0.63** | **−0.76, −0.49** | **<0.001** | **−0.95** | **−1.09, −0.81** | **<0.001** |
| Breast | **−0.85** | **−1.06, −0.64** | **<0.001** | | | |
| Prostate | | | | **−0.55** | **−0.74, −0.36** | **<0.001** |
| Lung and bronchus | **1.12** | **0.53, 1.85** | **<0.001** | −0.03 | −0.39, 0.33 | 0.887 |
| Colon and rectum | **−0.34** | **−0.59, −0.10** | **0.006** | **−0.79** | **−1.00, −0.59** | **<0.001** |
| Skin melanoma | **0.43** | **0.14, 0.72** | **0.004** | **−0.40** | **−0.70, −0.10** | **0.009** |
| Uterine corpus | −0.01 | −0.26, 0.24 | | | | |
| Bladder | | | | **0.53** | **0.24, 0.82** | **<0.001** |

Bold font indicates statistical significance at 0.05 level.

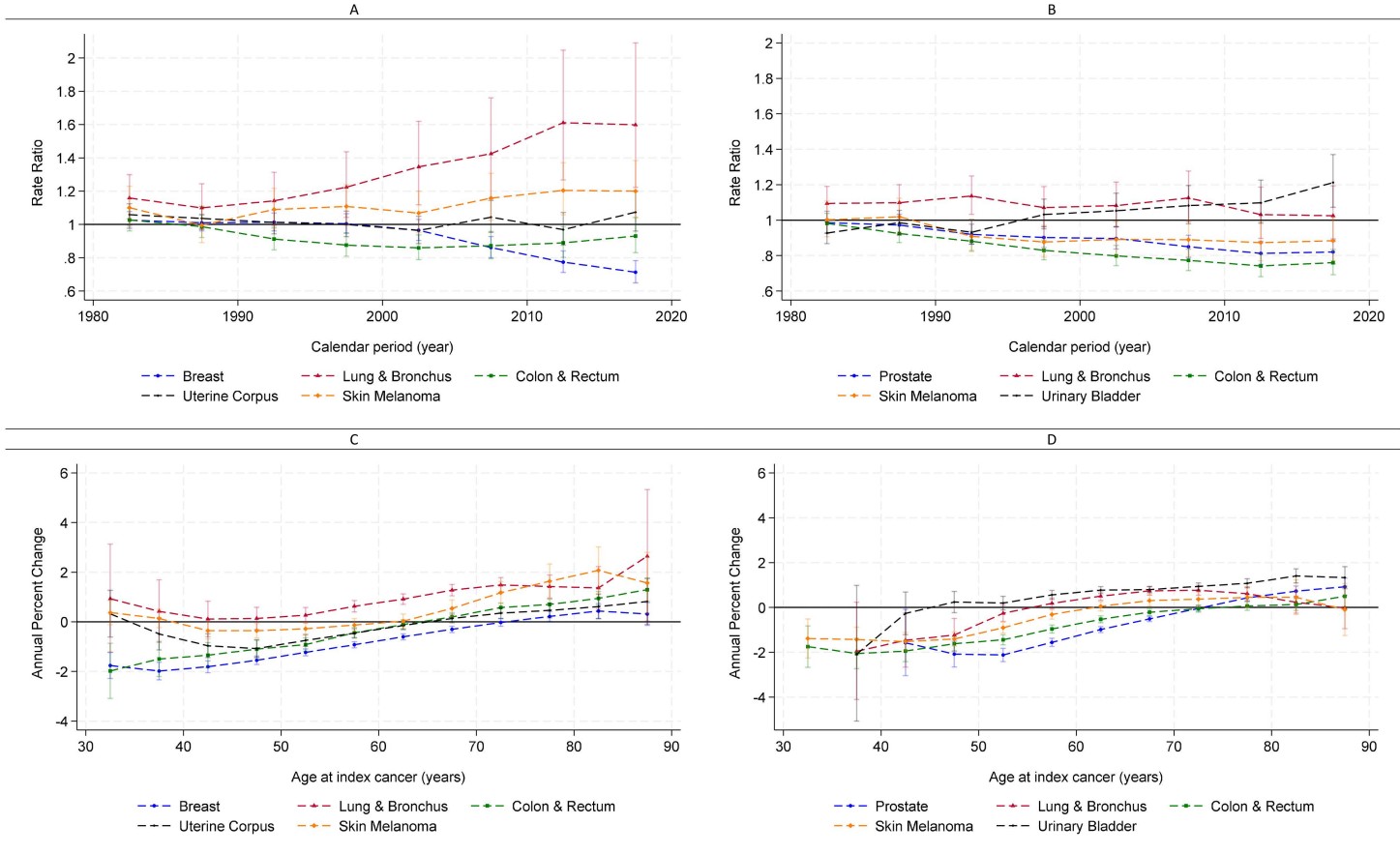

**Fig 4. Estimated period effect (rate ratio) and age-specific annual percent change of subsequent primary cancer (SPC) of subsequent primary cancer (SPC) by site of index primary cancer.** Data from SEER 1975-2019. **(A)** Period effect (rate ratio) among females. **(B)** Period effect (rate ratio) among males. **(C)** Age-specific annual percent change among females. **(D)** Age-specific annual percent change among males. Verticle bars represent 95% confidence intervals. For period effect, reference period = 1975–1980. A rate ratio >1 indicates higher incidence of SPC among cancer survivors diagnosed with the index cancer during a given period compared to index cancer diagnosed during 1975–1980 and vice versa. For annual percent change, a value >0 indicates an increasing trend, and a value <0 indicates a decreasing trend over time in SPC incidence among cancer survivors.

diagnosed with the index cancer at older age (Fig 4C and 4D). Specifically, increasing SPC incidence was observed among female survivors of lung cancer diagnosed after age 50 years, skin melanoma after age 65 years, colorectal and uterine corpus cancers after age 70 years, and breast cancer after age 75 years. Among male survivors, increasing SPC incidence was observed among those diagnosed with bladder cancer after age 55 years, lung cancer after age 60 years, and prostate cancer after age 75 years.

## Discussion

In this study, we conducted a large-scale age–period–cohort analysis to characterize temporal patterns in SPC risk among US cancer survivors. By disentangling the effects of biological aging, calendar period influences, and birth cohort exposures, this study reveals site- and sex-specific trajectories that have direct implications for survivorship care, cancer screening, and public health strategy.

SPC incidence increased with advancing age at diagnosis of the index cancer, consistent with cumulative exposure to carcinogenic factors, cumulative DNA damage, and immunosenescence [13,26–28]. The steep age-related increase in SPC risk among survivors of most index cancers highlights the importance of age-tailored surveillance strategies.

Men consistently exhibited higher SPC risks than women when the index cancer was diagnosed after age 20. These differences may reflect variations in biological susceptibility, lifestyle risk factors, and treatment exposures. Across sites of the index cancer, the higher SPC risks among survivors of lung and bronchus cancer, bladder cancer, and skin melanoma provides a basis to study the role of shared carcinogen exposures such as tobacco smoke and ultraviolet exposure [29–31].

In contrast to the steep incline of SPC risk with age at index cancer diagnosis, a stable SPC incidence was found among female breast cancer survivors. This pattern may reflect an age-dependent interplay of genetic, hormonal, and treatment-related factors. For example, genetic predisposition such as BRCA mutations may play a larger role in SPC development among breast cancer survivors diagnosed at a younger age [32,33]. Hormonal changes and senescence may play larger roles in SPC among survivors of later-onset breast cancer. Age-related differences in treatment intensity and modality (e.g., the dose and duration of hormonal therapy) can also contribute to risks of SPC [13,34–36]. These findings warrant further investigation into age-specific mechanisms underlying SPC risks in breast cancer survivors.

A noteworthy observation is that breast cancer (in women) and colorectal cancer (in men) diagnosed before age 40 years had the highest SPC risk among the most common index cancer sites studied. In addition to the well-documented role of germline predisposition in early-onset breast and colorectal cancers [33,37,38], recent studies suggest that obesity and sedentary lifestyle may contribute to rising incidence in younger populations [16,39]. Future studies are needed to examine the role of these factors in the development of SPCs. Nonetheless, our finding highlights the importance of integrating genetic counseling with lifestyle interventions into risk-stratified surveillance, particularly given that these cancers are among the fastest-growing malignancies in young Americans [25,35].

Cohort-specific SPC risks peaked in the 1935–1945 birth cohorts, suggesting that mid-20th-century behavioral and environmental exposures may have played a significant role in SPC development. Tobacco smoking, a major contributor to both first and subsequent primary malignancies [40], peaked in the 1960s in the US [41,42], when individuals born between 1935 and 1945 were 15–25 years old, an age range most vulnerable to smoking uptake [43]. In contrast, more recent birth cohorts may benefit from a range of public health improvements, including reduced smoking prevalence and enhanced food and occupational safety [37].

Nonetheless, SPC risks increased in more recent cohorts of female survivors of lung and bronchus cancer and male survivors of urinary bladder cancer without subsequent decline, and increases were observed among survivors of skin melanoma and prostate cancer in the most recent cohorts after declines in previous cohorts. These concerning patterns highlight the need for intensified surveillance and targeted prevention strategies (such as smoking cessation and lung cancer screening among female cancer survivors and dermatologic surveillance and ultraviolet protection strategies for male

skin melanoma survivors) for more recent cohorts of these cancers. Our findings provide a population-level foundation for such targeted investigations as well as studies on generational shifts in post-diagnosis risk factors, such as smoking, ultraviolet exposure, and sedentary lifestyle [44], to better understand and mitigate rising SPC risks in these cohorts.

Declines in SPC among the most recent cohorts of survivors of the most common cancers such as breast, prostate, and colorectal cancer, may reflect improvements in survivorship care such as lifestyle modification, genetic counseling, monitoring, and preventive treatment for these index cancers [35,45,46].

Encouragingly, SPC risk declined over calendar periods, particularly among survivors diagnosed at a younger age. These declines are consistent with improvements in cancer detection, diagnosis, and treatment, reduced exposure to carcinogens (such as pesticides, lead in gasoline and paint, and asbestos in household materials), and enhanced survivorship care [26,29,34,37,47,48]. Future studies are needed to assess the role of these factors. However, the increasing SPC risk among survivors diagnosed at an older age, especially those diagnosed after age 65 years, raises concerns about gaps in survivorship planning for aging populations. Adults 65 years of age and older account for 64% of cancer survivors in the US, and this number is projected to rise to 73% by 2040 [49]. Survivors diagnosed at an older age are often challenged by multiple comorbidities, which complicates survivorship care [50]. Our findings highlight the urgent need to integrate SPC surveillance and prevention into survivorship care among older adults. These findings highlight the need for future studies to test age-specific hypotheses about how clinical (e.g., treatment exposures and toxicity), behavioral (e.g., lifestyle changes), and policy factors (e.g., shifts in screening and survivorship guidelines) contribute to SPC risk in younger and older survivors to inform age-tailored prevention and surveillance strategies.

Site-specific patterns revealed divergent trajectories. Most alarming is the 60% increase of SPC incidence among female lung cancer survivors, and to a lesser extent, female skin melanoma survivors and male survivors of urinary bladder cancer. Prior studies indicate that subsequent primaries in lung and bladder cancer survivors often include smoking-related cancers [13,51], while subsequent primaries in melanoma survivors are dominated by melanomas, consistent with cumulative ultraviolet radiation exposure [52,53]. In addition, as survival continues to improve after a lung cancer diagnosis, a potential effect of treatment is SPC [54]. These findings underscore the need for future studies to clarify how behavioral factors (e.g., smoking intensity) and clinical factors (e.g., late-term toxicities from evolving thoracic treatments) may contribute to the rising SPC risk among female lung cancer survivors. Although our descriptive analyses cannot establish causality, these findings highlight the importance of tailoring surveillance and prevention strategies to the index cancer. For example, smoking cessation and lung/bladder cancer follow-up, ultraviolet protection, and dermatologic surveillance for melanoma survivors may represent opportunities to mitigate SPC burden. In addition, as the number of cancer survivors rises steeply due to population aging, better detection and diagnosis, and improved survival of the initial cancer [55], even stable or declining incidence can translate into a rising absolute burden of SPCs, placing increasing demands on surveillance systems, oncology services, and primary care.

Findings from this analysis of SPC risks among a large cohort of cancer survivors highlight the dynamic nature of SPC risk and the importance of incorporating age, sex, birth cohort, and index cancer site into survivorship care models. Our results provide essential surveillance evidence to identify survivor groups at greatest risk of SPC. These insights can help projection of future cancer burden and inform tailored survivorship care strategies, including risk-stratified screening, lifestyle interventions, and resource allocation. For example, the rising SPC incidence among female lung cancer survivors underscores the need for intensified surveillance programs and assessment of smoking status, while declines among breast cancer survivors highlight the potential impact of improved survivorship care. The increasing burden of SPCs among survivors diagnosed at older age and recent birth cohorts of female survivors of lung cancer and male survivors of urinary bladder cancer warrants renewed attention to long-term monitoring, risk-reducing strategies, and tailored screening guidelines. By delineating temporal patterns, our findings contribute to efforts aimed at reducing the future burden of SPCs and improving quality of life among cancer survivors. Future research should explore the underlying mechanisms driving these patterns and evaluate targeted interventions to mitigate SPC risk across diverse survivor populations. While

our analyses focused on SPC risk stratified by index cancer site, studies examining SPCs by site of the subsequent cancer can provide further insights about risk-based screening opportunities.

Several limitations should be noted when interpreting these results. First, while results from our study provided insights about emerging trends and can inspire hypothesis regarding the development of SPC, they do not link any specific exposure to SPCs. Because our age–period–cohort analyses are descriptive and each site-specific analysis involves a distinct survivor population, we did not apply formal multiplicity adjustments. Nonetheless, over-interpretation is possible given the large number of age-, period-, and cohort-strata. Second, we aimed to provide an overview of the SPC trends for most common index cancers, and therefore did not further stratify our analysis by subtypes of index cancers. More importantly, information about relevant risk factors is not available in the SEER data such as genetic disposition, treatment details, and lifestyle changes after the index cancer diagnosis. Future research is needed to understand the role of these factors in SPC to improve risk stratification and guide personalized survivorship care. Third, our analyses were restricted to 8 SEER registries. Individuals who migrated out of registry catchment areas were censored at the time of migration. Although incidence estimates remain valid for the follow-up period they contributed, SPCs diagnosed after migration would not be captured, which may reduce case counts and precision. Finally, SEER data, while high quality, can contain misclassified SPCs, particularly for histologically similar cancers at the same site [56]. Although we applied SEER multiple primary rules consistently [57], residual misclassification may persist. It is also possible that SPCs that occur but remain undiagnosed prior to death are not captured in SEER, which may lead to under-estimation of SPC incidence in certain populations. Improvements in diagnostic practices, coding, and registry coverage since 1975 may have also influenced SPC detection, particularly in the earlier years.

In this large, population-based study of cancer survivors spanning four decades, we used age–period–cohort analysis to delineate the distinct contributions of aging, calendar period, and generational variations to the incidence of SPCs. We found that SPC risk is shaped by complex, site-specific patterns, with some groups - such as female survivors of lung/bronchus cancer or skin melanoma and male survivors of bladder cancers - experiencing rising incidence of SPC despite overall population-level declines. These findings underscore the need for tailored, risk-stratified survivorship care informed by sex, age, the index cancer site, and temporal variations of SPC incidence. From a public health standpoint, anticipating future cancer burden will require greater integration of life span-based surveillance strategies, survivorship planning, and preventive care—particularly for survivors with rising period-associated risks.

Investments in scalable data infrastructure, generational risk assessment tools, and dynamic guideline development will be essential to support personalized, forward-looking survivorship care and to reduce preventable morbidity associated with multiple primary cancers.

## Supporting information

**S1 STROBE Checklist. Checklist of items that should be included in reports of cohort studies. This checklist is reproduced from the STROBE Statement (Strengthening the Reporting of Observational Studies in Epidemiology) and is licensed under the Creative Commons Attribution 4.0 International (CC BY 4.0).** *von Elm E, Altman DG, Egger M, Pocock SJ, Gøtzsche PC, Vandenbroucke JP; STROBE Initiative. The Strengthening the Reporting of Observational Studies in Epidemiology (STROBE)statement: guidelines for reporting observational studies. PLoS Med. 2007 Oct 16;4(10):e296. PMID: 17941714.*
(DOCX)

**S1 Supplemental Figures. Additional figures illustrating observed versus fitted incidence rates.** Fig A: Observed and fitted age-specific incidence among females. Fig B: Observed and fitted age-specific incidence among males.
(DOCX)

**S1 Table. Supplement—incidence of SPC: Counts of subsequent primary cancer events and person-years of follow-up for each 5-year age and calendar-year stratum; these serve as input data for the age–period–cohort**

**model.** Sheet 1. Overall data for males. Sheet 2. Overall data for females. Sheet 3. Data stratified by site of index cancer among males. Sheet 4. Data stratified by site of index cancer among females.
(XLSX)

**S2 Table. Supplement—APC estimates: Age–period–cohort model estimates, the numerical values underlying the figures.** Sheet 1: Estimated cohort rate ratio. Sheet 2: Estimated local drift. Sheet 3: Estimated longitudinal age pattern. Sheet 4: Estimated net drift. Sheet 5: Estimated period rate ratio.
(XLSX)

## Author contributions

**Conceptualization:** Hui G. Cheng, Oxana Palesh, Susan Hong.

**Data curation:** Hui G. Cheng.

**Formal analysis:** Hui G. Cheng.

**Investigation:** Hui G. Cheng, Chelsey McGill, Oxana Palesh, Susan Hong.

**Methodology:** Hui G. Cheng, Livingstone Aduse-Poku, Oxana Palesh, Susan Hong.

**Project administration:** Oxana Palesh, Susan Hong.

**Supervision:** Oxana Palesh, Susan Hong.

**Validation:** Livingstone Aduse-Poku.

**Visualization:** Hui G. Cheng.

**Writing – original draft:** Hui G. Cheng, Livingstone Aduse-Poku, Chelsey McGill.

**Writing – review & editing:** Oxana Palesh, Susan Hong.

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
