## [Editor Report · Decision Letter 0]

4 Oct 2025

Dear Dr Cheng,

Thank you for submitting your manuscript entitled “Trends in Subsequent Primary Cancer Incidence Among United States Cancer Survivors, 1975-2019: An Age-Period-Cohort Analysis” for consideration by PLOS Medicine.

Your manuscript has now been evaluated by the PLOS Medicine editorial staff as well as by an academic editor with relevant expertise and I am writing to let you know that we would like to send your submission out for external peer review.

However, before we can send your manuscript to reviewers, we need you to complete your submission by providing the metadata that is required for full assessment. To this end, please login to Editorial Manager where you will find the paper in the ‘Submissions Needing Revisions’ folder on your homepage. Please click ‘Revise Submission’ from the Action Links and complete all additional questions in the submission questionnaire.

For clinical studies, please upload a copy of your trial study protocol as a supporting information file. The study protocol should be the version submitted for approval to the institutional review board or ethics committee, should include any amendments to the study protocol, as well as the date of their approval by the institutional review or ethics committee. Please also detail any deviations from the study protocol in the Methods section of your manuscript. The editors will consider the protocol and study conduct prior to a final decision for external review.

Please re-submit your manuscript within two working days, i.e. by Oct 08 2025 11:59PM.

Kind regards,

Heather Van Epps, PhD

Consulting Editor

PLOS Medicine

---

## [Decision Letter · Decision Letter 1]

3 Dec 2025

Dear Dr Cheng,

Many thanks for submitting your manuscript “Trends in Subsequent Primary Cancer Incidence Among United States Cancer Survivors, 1975-2019: An Age-Period-Cohort Analysis” (PMEDICINE-D-25-03472R1) to PLOS Medicine. The paper has been reviewed by subject experts and a statistician; their comments are included below and can also be accessed here: [LINK]

As you will see, reviewers 2, 3 and 4 find the study of interest and find the report would be strengthened by additional information on the models and how the analyses were performed, discussion of study limitations and clarification of the advance provided by this manuscript. We further ask that you provide references to support speculation of mechanisms underlying changes in cancer recurrence over time. After discussing the paper with the editorial team and an academic editor with relevant expertise, I’m pleased to invite you to revise the paper in response to the reviewers’ comments. We plan to send the revised paper to some or all of the original reviewers, and we cannot provide any guarantees at this stage regarding publication.

We ask that you submit your revision by Dec 05 2025 11:59PM. However, if this deadline is not feasible, please contact me by email, and we can discuss a suitable alternative.

Don’t hesitate to contact me directly with any questions (atosun@plos.org).

Best regards,

Alison

Alison Farrell, PhD

Senior Editor

PLOS Medicine

on behalf of:

Alexandra Tosun, PhD

Senior Editor

PLOS Medicine

atosun@plos.org

Comments from the academic editor:

More details on the models and statistical analyses are required to better understand what the authors did.

Comments from the reviewers:

Reviewer #1: This study used a large sample size of more than 3 million cancer cases to study age, period and cohort effects on second primary cancer incidence by sex. Many of the analyses were conducted combining all initial sites and then repeated stratified on the initial site of the tumor. The stated gol is to improve quality of life and reduce cancer burden among cancer survivors, however, the link between the analyses and this stated goal is ambiguous. Another reasonable possible approach would be to identify patterns in second primary cancers by site of second cancer, particularly if one was interested in identifying risk-based screening opportunities. The interpretation of the site specific second primary data does not clearly lead to public health opportunities. Overall, some linguistic issues diminish the impact of this work. In addition, the author claims they have ‘disentangled’ age, period and cohort effects and this is unclear.

The statement in results (paragraph 1) that 'they represent 64% and 63% of all cancer survivors among women and men in the study period is not clear. Given that approximately 500K SPCs were detected among 3.7M survivors, these numbers seem incorrect, which suggests some specificity is lacking in this sentence. Similar ambiguity of phrasing occurs in referencing Figure 2 for the first time (Results paragraph 2), where authors state that age-related increases were observed among survivors of ‘the most index cancer sites’.

In describing the temporal trends, the authors describe ‘a bell curve’ which is a distribution within a population and not over time. This descriptor of the morphology of the curve is confusing. Similarly the term, n-shaped patterns was used, without definition.

The observation of lung, bladder and skin having patterns with age is attributed to tobacco smoke and UV exposure without justification (discussion).

Patterns of colorectal cancer and breast cancer in those diagnosed before 40 was cited as evidence of the need for early-life survivorship interventions (which seems unwarranted given the likely contribution of germline genetics to early onset breast and colorectal cancers, and possibly second primaries).

Declines over calendar period were attributed to detection, diagnosis, treatment, and reduced exposure and enhanced survivorship care. This speculative interpretation of the data does not substantially improve our appreciation for the dynamics of cancer prevention and control. In limitations, the authors acknowledge that the work inspires hypotheses (rather than clearly providing explanations), but the overall impact of the analyses remains unclear.

Visualization of the curves are small and not well-delineated with color and the site specific analyses are not interpreted in detail with respect to treatment-related exposures or other co-exposures. These gaps limit the ability of the paper to advance SPC prevention strategies.

Reviewer #2: OVERVIEW

This is an interesting and novel application of age-period-cohort analysis to characterize incidence rates of second primary cancers (SPCs) in an open cohort of cancer survivors. The cohort under observation is persons living in the SEER-8 catchment areas between 1975 - 2019, a population that includes roughly 10% of the total US population.

SEER-8 is the longest running nationally authoritative cancer registry in the US with almost 50 years of follow up. For this reason, it is optimal for obtaining a historical perspective on long-term cancer trends in the US, especially among non-Hispanic white men and women.

This historical perspective covers six social generations in the US. For adult cancers, from the Lost Generation through Generation X, and for childhood cancers, the Millennials.

The authors arrived at several useful take-home messages. Notably, they report that SPC risks increased in more recent cohorts of female survivors of lung and bronchus cancer and male survivors of urinary bladder cancer (Figure 3).

This report also highlights that much progress has been made, broadly speaking, since the peak of SPC susceptibility in persons born circa 1935 (Figure 1B). Having said that, it is also sobering to see evidence that the secular declines appear to have slowed since circa 2000 (Figure 1C).

MAIN COMMENTS

Basically, two suggestions.

The overall analysis approach is sound and demonstrates a nice application of what might be called ‘descriptive’ APC analysis.

The authors’ account of what they did is correct, but I would like to see more details about how the appropriate ‘Lexis diagrams’ were constructed using the SEER program data. (First suggestion).

This reviewer is not ‘hands-on’ when it comes to using SEER*Stat! Can the authors’ simply select second primary cancers only, and then conduct a rates session to obtain the appropriate tables of person-years at risk and corresponding numbers of events? Or, do you need to conduct a case listing session of cancer survivors so that these individuals can be censored at death? If the explanation is non-trivial, perhaps this material could be included in an online supplement. This might benefit other investigators.

Considering the longitudinal age incidence curves shown in Figure 1A, 2A, and 2B: These are very high rates. For example, in Figure 1A, above 2% per year among men over 60 and 1.5% per year among women over 60. To me this begs the question, how do the longitudinal age curves in Figure 1 compare with the corresponding longitudinal age curves for first malignancies obtained from the same SEER-8 population? (Second suggestions). Are we talking about a 50% increase, or more?

One reason for adding this information is it provides a nice quality-control check for the data assembly stage (first suggestion).

FWIW, such a comparative analysis technically describes the relative experience of the reference 1915 cohort, but in light of Figure 1B, these absolute curves are equally descriptive for the 1940 male birth cohort and the 1955 female cohorts.

This reviewer recognizes that extending such an analysis to every curve in Figure 2 goes beyond the scope of the present work, but perhaps merits mention as a possible

avenue for future studies.

Minor Comment:

Reference cohort in Fig 1 is also 1915, not 1885 as stated in the Figure 1 legend.

Philip S Rosenberg

Reviewer #3: Statistical review

This paper reports an analysis of prevalence of cancer recurrence and how this has changed over time, between the sexes and in different cancer types. I had some comments on the statistical methods and reporting, which are below.

1. Abstract: I am sympathetic about summarising a large number of results in an abstract. However, having some quantitative results for some of the statements would be useful, e.g. ‘increased with age..’, ‘steeper rises among men’.

2. “Cohort-specific SPC risk peaked in the 1935-1945 birth cohorts and declined in later cohorts, except among female survivors of lung cancer and male survivors of bladder cancer, where risks continued to rise.” - when I initially read this, it sounded like selected results from a large number of potential combinations. I don’t think it’s as bad as I initially thought due to there only being five cancer sites considered (although the number of combinations with age, sex etc is higher). Perhaps as a discussion point, were there any multiplicity concerns here?

3. Statistical analysis: it would be useful to mention that the modelling technique allows quantification of uncertainty (e.g. via confidence intervals).

4. Statistical analysis: my understanding of the models used is not good. Is this correct: a person’s follow-up would start at diagnosis of the index cancer and then the number of SPCs following is modelled as a Poisson outcome. Is the follow-up length included via an offset like would happen in a Poisson regression? If someone died of the index cancer, would the follow-up be truncated at death? Does this potentially ignore the potential for someone to have a SPC prior to death that is undiagnosed? I think including more intuition on the approach might be useful, together with any assumptions (and drawbacks in the discussion).

5. Statistical analysis: for the five different cancer sites, was this a stratified analysis using only data from people with that cancer site, or was a model with parameters for the cancer site used?

6. Statistical analysis: Because age, period, and cohort effects are linearly dependent (cohort = period − age), it might be helpful if the authors briefly note how the Rosenberg method addresses the identifiability issue and what constraints are imposed. This would reassure readers that the results reflect interpretable effects.

7. Results: Some indication of model fit (e.g. comparison of observed vs. fitted incidence rates, or summary of overdispersion parameters) would strengthen confidence in the results and clarify whether the APC model adequately captured the main features of the data.

James Wason

Reviewer #4: This is a very well written paper describing the risks of subsequent primary cancers (SPCs) among survivors of first primary cancers using data from 8 SEER registries. Strengths of the paper include the very large number of primary cancers (3.36 million) and >500,000 SPCs. This has allowed for robust analyses to investigate age, period and cohort effects in the 5 most common cancers in women and men. The results are presented clearly and are interpreted appropriately. Numbers of cancer survivors are growing around the world and warrant careful monitoring of risks and trends in SPCs. As the authors conclude, patterns of SPC risk are complex. The paper makes a valuable contribution to our understanding.

The data come from only 8 SEER registries which suggests that some primary and SPCs may have been missed if diagnosed in people who moved into or out of the areas covered by these registries. This limitation and its implications should be discussed. Discussion of the quality of the data and any changes in the study period from 1975 to 2022 would also be warranted

As a minor point, there is variable formatting of the references in the text (with or without a superscript). There is a typographical error in the footnote to table 1 (fond).

---

* Please upload any figures associated with your paper as individual TIF or EPS files with 300dpi resolution at resubmission; please read our figure guidelines for more information on our requirements: http://journals.plos.org/plosmedicine/s/figures. While revising your submission, we strongly recommend that you use PLOS’s NAAS tool (https://ngplosjournals.pagemajik.ai/artanalysis) to test your figure files. NAAS can convert your figure files to the TIFF file type and meet basic requirements (such as print size, resolution), or provide you with a report on issues that do not meet our requirements and that NAAS cannot fix.

After uploading your figures to PLOS’s NAAS tool - https://ngplosjournals.pagemajik.ai/artanalysis, NAAS will process the files provided and display the results in the “Uploaded Files” section of the page as the processing is complete.

If the uploaded figures meet our requirements (or NAAS is able to fix the files to meet our requirements), the figure will be marked as “fixed” above. If NAAS is unable to fix the files, a red “failed” label will appear above.

When NAAS has confirmed that the figure files meet our requirements, please download the file via the download option, and include these NAAS processed figure files when submitting your revised manuscript.

* Please ensure that the paper adheres to the PLOS Data Availability Policy (see http://journals.plos.org/plosmedicine/s/data-availability), which requires that all data underlying the study’s findings be provided in a repository or as Supporting Information. For data residing with a third party, authors are required to provide instructions with contact information (web or email address) for obtaining the data. Please note that a study author cannot be the contact person for the data. PLOS journals do not allow statements supported by “data not shown” or “unpublished results.” For such statements, authors must provide supporting data or cite public sources that include it.

* We expect all researchers with submissions to PLOS in which author-generated code underpins the findings in the manuscript to make all author-generated code available without restrictions upon publication of the work. In cases where code is central to the manuscript, we may require the code to be made available as a condition of publication. Authors are responsible for ensuring that the code is reusable and well documented. Please make any custom code available, either as part of your data deposition or as a supplementary file. Please add a sentence to your data availability statement regarding any code used in the study, e.g. “The code used in the analysis is available from Github [URL] and archived in Zenodo [DOI link]” Please review our guidelines at https://journals.plos.org/plosmedicine/s/materials-software-and-code-sharing and ensure that your code is shared in a way that follows best practice and facilitates reproducibility and reuse. Because Github depositions can be readily changed or deleted, we encourage you to make a permanent DOI’d copy (e.g. in Zenodo) and provide the URL.

* Please ensure that all COIs are disclosed, ethics statements are included if required, and a Data Availability Statement is included.

* Please clarify how no specific funding was involved. Are authors supported by grants?

* Please report author contributions and include an Acknowledgments section.

* Please ensure that the study is reported according to the appropriate guideline and include the completed checklist (e.g. STROBE) as Supporting Information. When completing the checklist, please use section and paragraph numbers, rather than page numbers. Please add the following statement, or similar, to the Methods: “This study is reported as per [XXXX] guideline (S1 Checklist).”

* At this stage, we ask that you include a short, non-technical Author Summary of your research to make findings accessible to a wide audience that includes both scientists and non-scientists. The Author Summary should immediately follow the Abstract in your revised manuscript. This text is subject to editorial change and should be distinct from the scientific abstract. Ideally each sub-heading should contain 2-3 single sentence, concise bullet points containing the most salient points from your study. In the final bullet point of ‘What Do These Findings Mean?’, please include the main limitations of the study in non-technical language. Please see our author guidelines for more information: https://journals.plos.org/plosmedicine/s/revising-your-manuscript#loc-author-summary.

* Please express the main results with 95% CIs as well as p values. When reporting p values please report as p<0.001 and where higher as the exact p value p=0.002, for example. Throughout, suggest reporting statistical information as follows to improve clarity for the reader “22% (95% CI [13%,28%]; p</=)”. Please be sure to define all numerical values at first use.

FIGURES AND TABLES

SUPPLEMENTARY MATERIAL

REFERENCES

[STUDY TYPE-SPECIFIC REQUESTS -

OBSERVATIONAL STUDIES

* Abstract: Please include the study design, population and setting, number of participants, years during which the study took place (enrollment and follow up), length of follow up, and main outcome measures.

* Please ensure that the study is reported according to the STROBE (or appropriate STOBE extension) guideline (available from: https://www.equator-network.org/reporting-guidelines/strobe) and include the completed STROBE (or STROBE extension) checklist as Supporting Information. Please add the following statement, or similar, to the Methods: “This study is reported as per the Strengthening the Reporting of Observational Studies in Epidemiology (STROBE) guideline (S1 Checklist).” When completing the checklist, please use section and paragraph numbers, rather than page numbers.

* [FOR POPULATION HEALTH/REGISTRY STUDIES] Please ensure that the study is reported according to the RECORD guideline (available from https://www.record-statement.org) and include the completed checklist as Supporting Information. Please add the following statement, or similar, to the Methods: “This study is reported as per the Reporting of Studies Conducted using Observational Routinely-Collected Data (RECORD) guideline (S1 Checklist).” When completing the checklist, please use section and paragraph numbers, rather than page numbers.

* [FOR POPULATION HEALTH ESTIMATES] Please ensure that the study is reported according to the GATHER statement (available from https://www.equator-network.org/reporting-guidelines/gather-statement) and include the completed checklist as Supporting Information. Please add the following statement, or similar, to the Methods: “This study is reported as per the Guidelines for Accurate and Transparent Health Estimates Reporting (GATHER) statement (S1 Checklist).” When completing the checklist, please use section and paragraph numbers, rather than page numbers.

* [FOR MEDIATION ANALYSES] We recommend that the study is reported according to the AGReMA statement (https://agrema-statement.org/#:~:text=AGReMA%20is%20an%20evidence%2D%20and,randomised%20trials%20and%20observational%20studies) and include the completed checklist as Supporting Information. Please add the following statement, or similar, to the Methods: “This study is reported as per the Guideline for Reporting Mediation Analyses (AGReMA) statement (S1 Checklist).” When completing the checklist, please use section and paragraph numbers, rather than page numbers.

* For all observational studies, in the manuscript text, please indicate: (1) the specific hypotheses you intended to test, (2) the analytical methods by which you planned to test them, (3) the analyses you actually performed, and (4) when reported analyses differ from those that were planned, transparent explanations for differences that affect the reliability of the study’s results. If a reported analysis was performed based on an interesting but unanticipated pattern in the data, please be clear that the analysis was data driven.

* Please state in the Methods section whether the study had a prospective protocol or analysis plan. If a prospective analysis plan (from your funding proposal, IRB or other ethics committee submission, study protocol, or other planning document written before analyzing the data) was used in designing the study, please include the relevant document(s) with your revised manuscript as a Supporting Information file to be published alongside your study and cite it in the Methods section. A legend for this file should be included at the end of your manuscript. If no such document exists, please make sure that the Methods section transparently describes when analyses were planned, and when/why any data-driven changes to analyses took place. Changes in the analysis, including those made in response to peer review comments, should be identified as such in the Methods section of the paper, with rationale.

MODELLING STUDIES

The following list is derived from Geoffrey P Garnett, Simon Cousens, Timothy B Hallett, Richard Steketee, Neff Walker. Mathematical models in the evaluation of health programmes. (2011) Lancet DOI:10.1016/S0140-6736(10)61505-X:

* If pertinent, please provide a diagram that shows the model structure, including how the natural history of the disease is represented, the process and determinants of disease acquisition, and how the putative intervention could affect the system.

* Please provide a complete list of model parameters, including clear and precise descriptions of the meaning of each parameter, together with the values or ranges for each, with justification or the primary source cited and important caveats about the use of these values noted.

* Please provide a clear statement about how the model was fitted to the data, including goodness-of-fit measure, the numerical algorithm used, which parameter varied, constraints imposed on parameter values, and starting conditions.

* For uncertainty analyses, please state the sources of uncertainties quantified and not quantified [can include parameter, data, and model structure].

* Please provide sensitivity analyses to identify which parameter values are most important in the model. Uncertainty estimates seek to derive a range of credible results on the basis of an exploration of the range of reasonable parameter values. The choice of method should be presented and justified.

* Please discuss the scientific rationale for the choice of model structure and identify points where this choice could influence conclusions drawn. Please also describe the strength of the scientific basis underlying the key model assumptions.

* For studies that develop a prediction model or evaluate its performance, please ensure that the study is reported according to the TRIPOD statement (https://www.equator-network.org/reporting-guidelines/tripod-statement) and include the completed checklist as Supporting Information. Please add the following statement, or similar, to the Methods: “This study is reported as per the Transparent Reporting of a Multivariable Prediction Model for Individual Prognosis Or Diagnosis (TRIPOD) statement (S1 Checklist).” For studies using machine learning, please use the TRIPOD-AI checklist. When completing the checklist, please use section and paragraph numbers, rather than page numbers.

---

## [Decision Letter · Decision Letter 2]

25 Feb 2026

Dear Dr. Cheng,

Thank you very much for re-submitting your manuscript “Trends in Subsequent Primary Cancer Incidence Among United States Cancer Survivors, 1975-2019: An Age-Period-Cohort Analysis” (PMEDICINE-D-25-03472R2) for review by PLOS Medicine.

Thank you for your detailed response to the reviewers’ and editors’ comments. I have discussed the paper with my colleagues and the academic editor, and it has also been seen again by two of the original reviewers. The changes made to the paper were satisfactory to the reviewers. As such, we intend to accept the paper for publication, pending your attention to the editors’ comments below in a further revision. When submitting your revised paper, please once again include a detailed point-by-point response to the editorial comments. The remaining issues that need to be addressed are listed at the end of this email.

In revising the manuscript for further consideration here, please ensure you address the specific points made by the editors. In your rebuttal letter you should indicate your response to the editors’ comments and the changes you have made in the manuscript. Please submit a clean version of the paper as the main article file. A version with changes marked must also be uploaded as a marked up manuscript file. Please also check the guidelines for revised papers at http://journals.plos.org/plosmedicine/s/revising-your-manuscript for any that apply to your paper.

Please note, when your manuscript is accepted, an uncorrected proof of your manuscript will be published online ahead of the final version, unless you’ve already opted out via the online submission form. If, for any reason, you do not want an earlier version of your manuscript published online or are unsure if you have already indicated as such, please let the journal staff know immediately at plosmedicine@plos.org.

We ask that you submit your revision by Mar 04 2026. However, if this deadline is not feasible, please contact me (atosun@plos.org) or the journal staff by email, and we can discuss a suitable alternative.

We look forward to receiving the revised manuscript.

Sincerely,

Alexandra Tosun, PhD

Senior Editor

PLOS Medicine

plosmedicine.org

Comments from Reviewers:

Reviewer #2: The authors thoroughly responded to all of my suggestions. I have no further comments.

Reviewer #3: Thank you to the authors for addressing all my previous comments well. I have no further issues to raise.

Comments from the academic editor:

I believe the authors have done a great work and have adequately addressed all comments and concerns raised by the 4 reviewers. In my opinion, this is a very interesting and high-quality manuscript, with well-performed analyses and good presentation and interpretation of results.

Requests from Editors:

GENERAL

* Please confirm that your title complies with to PLOS Medicine’s style. Your title must be nondeclarative and not a question. It should begin with main concept if possible. “Effect of” (or “Impact of”) should be used only if causality can be inferred, i.e., for an RCT. Please place the study design (“A randomized controlled trial,” “A retrospective study,” “A modelling study,” etc.) in the subtitle (ie, after a colon).

* Statistical reporting: Please revise throughout the manuscript, including tables and figures.

- Please report statistical information as follows to improve clarity for the reader, ““XX% (95% CI [XX,YY]; p</=)””.

- Please separate upper and lower bounds with commas instead of hyphens as the latter can be confused with reporting of negative values.

- Please repeat statistical definitions (HR, CI etc.) for each set of parentheses.

* Please ensure that all abbreviations are defined at first use throughout the text (including statistical abbreviations).

* Please ensure that tables and figures, including those in supplementary files, are appropriately referenced in the main text.

* Please review your text for claims of novelty or primacy (e.g. ‘for the first time’ or ‘novel’) and remove this language.

* Please confirm that any use of statistical terms (such as trend or significant) are supported by the data, and if not please remove them. The term trend should be used only when the test for trend has been conducted.

* Please define all acronyms used in each figure or table in the corresponding legend.

* Please confirm that you used patient-centered language. Please note that patient-centered language is constructed with the use of post-modified nouns putting the person first in the sentence structure.

* When reporting age, please include ‘years’ as unit.

* Data Availability Statement: Please note that the data sharing statement following the main text and the details in online submission from do not match. Please revise. Could provide more details on the exact datasets you used in your study from SEER research data?

* Please explain why the Supplemental incidence tables are for review purposes only? If these tables provide the number behind the graphs, we require you to make these available as part of the publication.

ABSTRACT

* Please confirm that your abstract complies with our requirements, including providing all the information relevant to this study type https://journals.plos.org/plosmedicine/s/submission-guidelines#loc-abstract

* Please ensure to quantify the main results (with 95% CIs and p values).

* Please confirm that all numbers presented in the abstract are present and identical to numbers presented in the main manuscript text.

* In the abstract, please confirm you included the important dependent variables that are adjusted for in the analyses.

* In the last sentence of the Abstract Methods and Findings section, please describe the main limitation(s) of the study’s methodology.

METHODS AND RESULTS

* Please remove the numbering from the headings.

* Following the comment above, please revise the completed STROBE checklist and use section and paragraph numbers, e.g. “Methods, Paragraph 2”.

* “Input and output files are available upon request.” – please note that PLOS Medicine requires that the de-identified data underlying the specific results in a published article be made available, without restrictions on access, in a public repository or as Supporting Information at the time of article publication, provided it is legal and ethical to do so.

a) If the data are owned by a third party but freely available upon request, please note this and state the owner of the data set and contact information for data requests (web or email address). Note that a study author cannot be the contact person for the data.

b) If the data are not freely available, please describe briefly the ethical, legal, or contractual restriction that prevents you from sharing it. Please also include an appropriate contact (web or email address) for inquiries (again, this cannot be a study author).

* Figure 1 requires revision:

a) We suggest specifying the x-axis label, e.g. ‘Age at index cancer (years)’.

b) We suggest specifying the x-axis label to ‘Cohort (Birth year)’ (if correct).

c) Please specify how ‘Period’ is defined.

d) Please add a unit where missing.

* Please revise the figures according to the above comments and ensure to provide specific labels and descriptions.

* “Observed age‑specific incidence and fitted longitudinal curves were shown in Supplemental Figures” – please provide a specific reference, e.g. Figures S1 and S2.

* Figure 1B: It looks like for women the decline plateaued between 1980 and 1985 before continuing to decline.

* “Across sites of the index cancer, uterine corpus cancer survivors had the lowest SPC incidence among women, and prostate cancer survivors had the lowest incidence among men.” – At all ages? The statement doesn’t seem to be 100% accurate.

* Please confirm that where relevant figures include 95% CIs.

* Please confirm that you specified the variables controlled for in all relevant Tables.

* We still feel that the visualization of the curves could be improved, as overlapping curves are sometimes difficult to distinguish. Also, please ensure that the values behind the graphs are available in the supplementary tables.

DISCUSSION

* Please remove the ‘conclusions’ subheading from the discussion. Please also remove any other subheadings from the discussion.

* “Cohort-specific SPC risks followed a bell-shaped trajectory” – Please refer to the previous comment from Reviewer #1. In your response, you indicated that you revised any such mentions.

* “We also strengthened the framing of our contribution: our analyses provide population-level surveillance evidence that identifies survivor groups with rising or declining SPC risks, thereby informing hypotheses for future analytic studies and guiding resource allocation in survivorship care (Discussion section).” – iIt seems that you have not framed your work as inspiring hypotheses, as the reviewer suggested. Please comment.

General Journal Requests

2) Please ensure that the paper adheres to the PLOS Data Availability Policy (see http://journals.plos.org/plosmedicine/s/data-availability), which requires that all data underlying the study’s findings be provided in a repository or as Supporting Information. For data residing with a third party, authors are required to provide instructions with contact information for obtaining the data. PLOS journals do not allow statements supported by “data not shown” or “unpublished results.” For such statements, authors must provide supporting data or cite public sources that include it.

---

## [Editor Report · Decision Letter 3]

19 Mar 2026

Dear Dr Cheng,

On behalf of my colleagues and the Academic Editor, Mahdi Sheikh, I am pleased to inform you that we have agreed to publish your manuscript “Subsequent Primary Cancer Incidence Among United States Cancer Survivors, 1975-2019: An Age-Period-Cohort Analysis” (PMEDICINE-D-25-03472R3) in PLOS Medicine.

I appreciate your thorough responses to the reviewers’ and editors’ comments throughout the editorial process. We look forward to publishing your manuscript, and editorially there are only a few remaining points that should be addressed prior to publication. We will carefully check whether the changes have been made. If you have any questions or concerns regarding these final requests, please feel free to contact me at atosun@plos.org.

Please see below the minor points that we request you respond to:

* We suggest changing the title to: Subsequent Primary Cancer Incidence Among Cancer Survivors in the United States, 1975-2019: An Age-Period-Cohort Analysis

* Data Availability Statement: We have updated the DAS to include the following statement: “Incidence data and age-period-cohort model estimates are provided in Supporting Information.” Please let us know whether you agree with this change.

* Please include a reference to the STROBE checklist in the main text.

Before your manuscript can be formally accepted you will need to complete some formatting changes, which you will receive in a follow up email (including the editorial requests above). Please be aware that it may take several days for you to receive this email; during this time no action is required by you. Once you have received these formatting requests, please note that your manuscript will not be scheduled for publication until you have made the required changes.

In the meantime, please log into Editorial Manager at http://www.editorialmanager.com/pmedicine/, click the “Update My Information” link at the top of the page, and update your user information to ensure an efficient production process.

PRESS

Sincerely,

Alexandra Tosun, PhD

Senior Editor

PLOS Medicine